# The Heterogeneity of the Tumor Microenvironment as Essential Determinant of Development, Progression and Therapy Response of Pancreatic Cancer

**DOI:** 10.3390/cancers13194932

**Published:** 2021-09-30

**Authors:** Anna Maxi Wandmacher, Anne-Sophie Mehdorn, Susanne Sebens

**Affiliations:** 1Institute for Experimental Cancer Research, Kiel University and University Hospital Schleswig-Holstein Campus Kiel, Arnold-Heller-Str. 3, Building U30 Entrance 1, 24105 Kiel, Germany; 2Department of Internal Medicine II, University Hospital Schleswig-Holstein Campus Kiel, Arnold-Heller-Str. 3, 24105 Kiel, Germany; annamaxi.wandmacher@uksh.de; 3Department of General, Visceral, Thoracic, Transplantation and Pediatric Surgery, University Hospital Schleswig-Holstein Campus Kiel, Arnold-Heller-Str. 3, Building C, 24105 Kiel, Germany; Anne-Sophie.Mehdorn@uksh.de

**Keywords:** PDAC, tumor stroma, immune cells, endothelial cells, CAFs, myofibroblasts, microbiome, macrophages, personalized medicine

## Abstract

**Simple Summary:**

Pancreatic ductal adenocarcinoma (PDAC) is still one of the deadliest cancers in western countries. It is commonly diagnosed at advanced stages and most anti-cancer therapies have failed to substantially improve prognosis of PDAC patients. PDAC is characterized by a profound inflammatory tumor microenvironment (TME) comprising various non-neoplastic cells e.g., myofibroblasts, macrophages, T cells and endothelial cells which can exhibit different functional phenotypes. Furthermore, the microbiome is altered in the tumor and other body compartments of PDAC patients adding to the great TME heterogeneity and its impact on PDAC development, progression and therapy responses. This review summarizes the recent knowledge on the diverse phenotypes of these different stromal components. A better understanding of tumor cells as well as TME heterogeneity and considering tumor-suppressing and tumor-promoting phenotypes might provide an important step towards a more effective treatment for this highly malignant tumor.

**Abstract:**

Pancreatic ductal adenocarcinoma (PDAC) is commonly diagnosed at advanced stages and most anti-cancer therapies have failed to substantially improve prognosis of PDAC patients. As a result, PDAC is still one of the deadliest tumors. Tumor heterogeneity, manifesting at multiple levels, provides a conclusive explanation for divergent survival times and therapy responses of PDAC patients. Besides tumor cell heterogeneity, PDAC is characterized by a pronounced inflammatory stroma comprising various non-neoplastic cells such as myofibroblasts, endothelial cells and different leukocyte populations which enrich in the tumor microenvironment (TME) during pancreatic tumorigenesis. Thus, the stromal compartment also displays a high temporal and spatial heterogeneity accounting for diverse effects on the development, progression and therapy responses of PDAC. Adding to this heterogeneity and the impact of the TME, the microbiome of PDAC patients is considerably altered. Understanding this multi-level heterogeneity and considering it for the development of novel therapeutic concepts might finally improve the dismal situation of PDAC patients. Here, we outline the current knowledge on PDAC cell heterogeneity focusing on different stromal cell populations and outline their impact on PDAC progression and therapy resistance. Based on this information, we propose some novel concepts for treatment of PDAC patients.

## 1. Clinical Situation and Challenges

Pancreatic ductal adenocarcinoma (PDAC) is the 4th most frequent cause of cancer related deaths in western countries [1]. Since 80% of PDAC are diagnosed at advanced tumor stages, curative treatment options are limited. Accordingly, the overall 5-year survival rate is still less than 10% [1]. Besides late diagnosis, an early metastatic dissemination as well as a profound therapy resistance contribute to the clinical challenges in the treatment of PDAC and thereby to the patient’s dismal situation [2,3].

Originating from pancreatic ductal epithelial cells (PDEC) or pancreatic acinar cells, PDAC can develop via different precursor lesions, such as pancreatic intraepithelial neoplasia (PanIN), intraductal papillary mucinous neoplasm (IPMN), mucinous cystic neoplasm, or other lesions [4], to an invasive carcinoma with the liver being the main site of metastasis [5,6]. During this multi-step progression various epi-/genetic alterations are acquired e.g., mutations in the oncogene *KRAS* which is one of the earliest genetic alterations and found in 99% of even early low-grade PanIN [7,8]. Besides genetic and epigenetic changes in PDEC and later on PDAC cells, the adjacent microenvironment also undergoes considerable alterations. These involve enrichment of different leukocyte cell populations (e.g., monocytes, lymphocytes), activation and modification of tissue resident cells (e.g., fibroblasts/stellate cells, endothelial cells), remodeling of the extracellular matrix (ECM) and changes of the microbiome [9,10,11]. Accordingly, composition and proportion of the microenvironment in close vicinity to precursor/PDAC cells are highly variable as they evolve concomitantly with PDAC progression. Furthermore, the TME is mutually dependent on both environmental factors (e.g., aging, exposure to lifestyle factors) and the (epi-)genetic make-up of PDEC and PDAC cells, respectively [12]. This indicates a dynamic co-evolution of neoplastic and non-neoplastic cells during pancreatic tumorigenesis being characterized and driven by a high heterogeneity of both different tumor cell clones and stromal cell populations. PDAC is characterized by a pronounced inflammatory tumor microenvironment (TME), which often accounts for the major tumor mass. Therefore, it is not surprising that therapies targeting genetically related alterations (e.g., altered signaling by mutant *KRAS* or sustained Epidermal Growth Factor Receptor (EGFR)-signaling caused by *EGFR* overexpression) have failed to improve treatment of PDAC patients as the blocked signaling pathways are bypassed by the stroma-mediated signaling which further sustains the cellular processes in the transformed cells [13,14]. Similarly, chemotherapeutic regimens have largely failed to improve the prognosis of PDAC patients because PDAC cells exhibit multiple strategies by which they evade the effect of cytostatic drugs. Importantly, these strategies are promoted by the stromal cells and additionally, the TME itself impairs the drug efficacy of PDAC (e.g., by metabolization of cytostatic drugs) [15,16,17,18]. In this context, the microbiome seems to play an important role, too. Several studies have already shown that the gut and tumor microbiome are altered in PDAC patients, promoting tumor progression and reducing treatment responses of PDAC cells, thereby substantially impairing survival of PDAC patients [11,19,20].

Finally, immune checkpoint inhibitors have revolutionized the treatment of various solid tumors even at advanced stages such as non-small cell lung carcinoma but have failed in the treatment of PDAC patients [21,22,23]. Besides a heterogeneous expression pattern of the immune checkpoint regulator programmed cell death 1 ligand 1 (PD-L1) on PDAC cells [24], another reason might be the high heterogeneity of the stromal compartment not only with respect to PD-L1 expression, but also regarding the cellular composition (inflamed versus non-inflamed tumors) [25,26].

Thus, experimental and clinical data strongly support the view that the failure of current therapeutic strategies is due to an insufficient consideration of the heterogeneity of PDAC cells but also stromal cells. In order to improve our understanding of this heterogeneity, we herein outline the recent knowledge on TME diversity and its impact on PDAC development, progression and therapy resistance. Being aware of the numerous publications in this field in recent years, we apologize for only citing a selection. Furthermore, considering this knowledge we propose some concepts for novel therapeutic strategies to improve prognosis of PDAC patients.

## 2. Heterogeneity of the Tumor Cell Compartment in PDAC

Comprehensive (single cell) omics analyses revealed a high diversity of tumors at multiple levels, namely interpatient, intertumor and intratumor heterogeneity altogether providing an explanation for the noticeable differences in therapy responses and survival of cancer patients [27]. In stroma-rich PDAC, those analyses were combined with microdissection allowing a distinct investigation of tumor and stromal cells. In recent years, mainly three studies revealed substantial insights into interpatient heterogeneity. Collisson et al. classified PDAC into “classical”, “quasi-mesenchymal” and “exocrine-like” subtypes. Besides differences in the expression profile, the classical phenotype was an independent prognostic factor for longer overall survival of resected PDAC patients and classical PDAC lines were more resistant towards Gemcitabine treatment than cell lines with a quasi-mesenchymal phenotype [28].

Moffitt et al. identified two tumor cell subtypes (“basal-like” and “classical”, the latter one overlapping with those identified by Collisson et al.) but additionally two stromal subtypes (“normal” and “activated”). Here, patients with a tumor of the “basal-like” subtype or exhibiting an activated stroma had a worse prognosis compared to either corresponding subgroup. However, basal-like tumors seemed to show better responses towards adjuvant therapy. Importantly, the normal stroma signature was associated with high expression of genes typical for pancreatic stellate cells (PSC), while an activated stroma signature was characterized by high expression of genes related to macrophages and myofibroblasts, both being abundant in the PDAC stroma [29].

Finally, Bailey et al. identified four subtypes, namely “squamous”, “pancreatic progenitor”, “immunogenic” and “aberrantly differentiated endocrine exocrine (ADEX)”. These subtypes clearly differed with respect to genetic and epigenetic alterations as well as transcription profiles. Additionally, the “immunogenic” subtype was characterized by an enrichment of different immune cell subsets and enhanced immune evasion pathways e.g., mediated by the immune checkpoint regulators Cytotoxic T-Lymphocyte-Associated Protein 4 (CTLA-4) and Programmed cell death protein 1 (PD-1) [30].

Extending these findings, Birnbaum et al. compared the tumor subtypes identified by Collisson, Moffitt and Bailey and confirmed the independent prognostic value of the classification from the latter two studies [31]. Additionally, Rashid et al. provided evidence that a tumor classification into two subtypes (basal-like and classical) according to Moffitt et al. exhibits the highest robustness and clinical relevance by demonstrating that basal-like tumors showed a strong resistance to FOLFIRINOX compared to classical tumors [32]. Although all studies revealed important insights into interpatient tumor heterogeneity, prospective trails are still needed to validate the clinical utility of this tumoral subtyping in particular with respect to optimized treatment.

With respect to intratumor heterogeneity, it is assumed that phenotypic and genotypic diversity of tumor cell clones within (primary and secondary) tumors originate from clonal expansions and differentiation hierarchies according to the current model of cancer stem cells (CSC) [33,34]. Thus, intratumor heterogeneity can occur and manifest on multiple levels based on the expansion of tumor cell clones with different genetic (driver as well as passenger mutations) and epigenetic alterations (hypermethylated DNA regions, histone modifications) as well as divergent differentiation stages (CSC versus non-CSC stage). These alterations in turn essentially contribute to different metabolic requirements (dependence on glucose, lactate or glutamine) as well as phenotypic diversity within the tumor mass resulting in e.g., proliferating versus resting, sessile versus motile/invasive or therapy responsive or resistant tumor cell clones [35,36]. Important to note, expansion and evolution of certain tumor cell clones are highly dependent on the adjacent stroma which co-evolves with the different tumor cell clones during pancreatic tumorigenesis, applying to the primary as well as to the secondary context [37,38]. Although it can be speculated whether the metastatic tumor stroma induces the above-mentioned epigenetic, metabolic and thereby phenotypic alterations or selects for those clones that have acquired these alterations before leaving the primary tumor, a fundamental role of the stromal compartment in this heterogeneity determining process is meanwhile indisputable.

Thus, it is necessary to unravel all levels of tumor diversity and in particular stromal cell heterogeneity (Figure 1) to gain a complete picture of PDAC complexity which has to be considered for optimized treatment strategies.

## 3. Heterogeneity of the Stromal Compartment in PDAC

### 3.1. Heterogeneity of Carcinoma Associated Fibroblasts

The most abundant inflammatory stroma cells in PDAC are myofibroblasts, also termed carcinoma associated fibroblasts (CAF). CAF are abundant in primary PDAC as well as in metastases e.g., in the liver. During cancer development, several cell types can transdifferentiate into CAF. Besides PSC, adipocytes, monocytes, bone marrow-derived mesenchymal stem cells and endothelial cells (EC) can give rise to CAF in the pancreas [39,40,41,42,43]. In the liver, the main source of CAF is hepatic stellate cells (HSC), with CAF also originating from hepatic sinusoidal endothelial cells or portal mesenchymal cells [44,45,46,47,48,49,50].

Under physiological conditions, most pancreatic fibroblasts are a heterogeneous group of desmin+ PSC which produce and maintain physiological levels of ECM proteins such as collagen IV, laminin, fibronectin and glycosaminoglycans. Thereby, these cells essentially contribute to the physiological pancreas homeostasis [51,52,53,54,55,56]. PSC are located at the base of the acini and contain high amounts of vitamin A vesicles [51]. Upon activation PSC lose these lipid vesicles and transdifferentiate into myofibroblasts, thereby increasing their proliferative and migratory potential, suppressing pro-apoptotic gene expression and secreting cell cycle promoting proteins [57,58,59]. Furthermore, myofibroblasts produce elevated amounts of ECM proteins at an altered composition, resulting in fibrosis. This fibrotic ECM is composed of less elastin and more fibronectin as well as crosslinked collagen type I and III [60]. Baron et al. distinguished two groups of pancreatic myofibroblasts: The first phenotype is characterized by production of ECM, and the second phenotype, called immune-activated PSC, exhibits and releases high expression levels of immunomodulating cytokines, interleukins and chemokines (e.g., chemokine (C-X-C motif) ligand 3 (CXCL3), Interleukin-33 (IL-33), IL-2) [61]. Exposure to stressful conditions including exogenous toxins, surgical intervention, inflammatory injury or mediators released by immune cells [56,62,63] promotes proliferation and activation of PSC into myofibroblasts leading to increased alpha-smooth muscle actin (α-SMA) expression. Stimuli leading to the transdifferentiation and activation of myofibroblasts/CAF are Transforming Growth Factor-beta (TGF-β), Platelet-derived Growth Factor (PDGF) and IL-6 [64,65,66,67]. Meanwhile, it is well known that CAF can exhibit different phenotypes and different CAF populations have been discovered in recent years. In the pancreas, major subgroups are myofibroblastic CAF (myCAF), inflammatory CAF (iCAF) and major histocompatibility complex II (MHCII) antigen presenting CAF (apCAF) [68]. MyCAF, which are tumor-adjacent and express high levels of α-SMA, are activated in a TGF-β dependent manner [69,70]. In contrast, iCAF are localized distant from the tumor, express low α-SMA levels and exhibit rather tumor-promoting and immunosuppressive properties by secreting inflammatory cytokines, like IL-6, CXCL12 and Granulocyte-Colony stimulating Factor (G-CSF). In contrast to myCAF, iCAF differentiation is stimulated by cancer-secreted IL-1 via the Januskinase-Signal Transducer and Activator of Transcription (JAK-STAT) pathway and antagonized by TGF-β. In vitro, PSC can differentiate into either myCAF or iCAF and even after this differentiation, iCAF can transdifferentiate into tumor-suppressive myCAF [68,69]. These findings strongly support the existence of different (functional) CAF phenotypes which seem to be interchangeable and highly context dependent. In both murine and human tissues, Hutton et al. recently identified two further functionally distinct pancreatic fibroblast populations by cluster of differentiation 105 (CD105) expression. While CD105+ fibroblasts did hardly impact tumor growth in vivo, CD105- fibroblasts promoted anti-tumor immunity and suppressed tumor growth. Characteristics of the established myCAF and iCAF populations were found in either lineage, whereas all apCAF were CD105-. In contrast to myCAF and iCAF populations, CD105+ and CD105- pancreatic fibroblasts seemed not to be interconvertible [71].

As outlined above, CAF are the major source of ECM in the TME. Cross-linked and oriented collagens cause a stiffened matrix, which stimulates accelerated cell cycle progression in adjacent tumor cells and thus fuels tumor growth [72]. Besides, the dense stroma acts as a physical barrier for drugs and immune cells [69]. Neumann et al. showed that the response to chemotherapeutic treatment (e.g., Gemcitabine, Nab-paclitaxel) is reduced when tumor cells are directly cocultured with human CAF compared to monocultured tumor cells [73]. Furthermore, T cells are unable to infiltrate into the dense matrix, which might be one reason for the limited clinical success of immunotherapy in PDAC patients [74]. By compressing blood and lymphatic vessels, perfusion is impaired and the interstitial fluid pressure increased, which induces hypoxia mediated progression and promotes cancer invasiveness, respectively [75,76,77,78]. Hypoxia in turn selects the most malignant cancer cells and through hypoxia-inducible factor 1 (HIF-1) suppresses tumor suppressor genes while promoting the metastatic cascade [79,80,81,82]. By promoting M2-like polarization in macrophages and increasing PD-L1 expression in cancer cells, tumor-associated macrophages (TAM) and dendritic cells (DC), HIF-1 supports immunosuppression in the TME [83,84,85,86,87].

Besides the ECM-mediated effects, CAF exert several direct effects on the tumor. Thus, CAF promote an immunosuppressive TME by selecting tumor-promoting immune cells and inhibiting tumor-suppressive ones. Hence, in a KPC mouse model, increased PD-L1 and PD-L2 expression was demonstrated on CAF compared to fibroblasts [88]. Multiple murine and human studies showed their capability of excluding cytotoxic T cells (CTL) from tumor islands and inhibiting T cell activity in multifaceted ways [88,89,90,91]. In a mouse model, Feig et al. found CAF-derived CXCL12 to coat PDAC cells and protect them from CD8+ T cell invasion [90]. Lakins et al. showed murine CAF to cross-present tumor-derived antigens on MHC I and upon contact with CTL, induce cell death via PD-1/PD-L2 and Fas/FasL interaction [91]. While preventing CTL functions, CAF recruit immunosuppressive immune cells, such as myeloid derived suppressor cells (MDSCs) and monocytes, and promote monocyte differentiation into M2-like macrophages via Macrophage-Colony Stimulating Factor (M-CSF) [92,93,94,95,96]. MDSC-promoting factors secreted by CAF include IL-6, Vascular Endothelial Growth Factor (VEGF) and M-CSF [97]. In PDAC, iCAF have been identified to be the major source of these immunomodulating ligands, which might explain their above-mentioned localization at the edge of the tumor [24,70,98,99,100]. Another CAF subpopulation found in murine and human PDAC and breast cancer are apCAF [101], which are characterized by expression of MHC II molecules and their capability to present antigens to CD4+ T cells. However, apCAF lack costimulatory molecules required to promote T cell proliferation. Thus, MHC II molecules on apCAF were hypothesized to act as decoy receptors and exert immunosuppression via induction of anergy in T cells or transdifferentiation into regulatory T cells (Tregs) [100]. Furthermore, CAF can enhance cell growth [102], induce Epithelial-Mesenchymal-Transition (EMT) thereby facilitating PDAC cell dissemination and metastasis and promote an apoptosis and drug resistant phenotype in PDAC as well as precursor cells in a paracrine manner [17,18,103,104].

The mechanisms of myofibroblasts/CAF generation and their functional impact on tumorigenesis apply also to HSC and the derived hepatic myofibroblasts (HMF) [105,106,107]. HSC make up 5% of all liver cells and share many properties with PSC [108,109]. In the physiological liver, HSC perform a variety of tasks to maintain organ homeostasis by paracrine secretion of hepatocyte mitogens such as Hepatocyte Growth Factor (HGF) and Epidermal Growth Factor (EGF), and they store 50–80% of all retinoids in the body [110,111,112,113,114,115]. In contrast in the inflamed liver, HMF essentially impact immune responses inducing infiltration of neutrophils and lymphocytes or by acting as professional antigen presenting cells (APC), while also inhibiting T cells via the expression of PD-L1 [116,117,118,119,120]. In a KPC mouse model, Costa-Silva et al. [105] demonstrated that PDAC-derived circulating exosomes are taken up by Kupffer cells inducing TGF-β expression in these liver resident macrophages. TGF-β in turn activates HSC to HMF leading to an increased production of fibronectin and collagen I and III. This pre-metastatic niche was shown to be crucial for the onset of PDAC metastases [105,121,122,123,124]. Recently, Bhattacharjee et al. also identified different CAF populations in murine and human liver tissues based on subpopulations previously characterized in pancreatic and breast cancer. Thus, most liver CAF exhibit a myCAF-like expression profile with high collagen and α-SMA levels, while a smaller subpopulation expresses less α-SMA, but more growth factors and cytokines, and is therefore defined as iCAF [125]. Besides the role in immunomodulation and ECM remodeling, HSC and HMF differentially impact cell growth of PDAC cells, thereby controlling outgrowth of liver metastases. Accordingly, Lenk et al. showed that both murine and human HSC inhibit PDEC and promote a dormant phenotype in an IL-8 dependent manner, while HMF promote reawakening of the dormant stage and enhance proliferation in a VEGF dependent fashion [106].

### 3.2. Approaches for Targeted Therapy of Carcinoma Associated Fibroblasts

In light of the broad impact of CAF on PDAC progression, immune modulation and drug response, targeting CAF, CAF-derived factors or factors leading to their generation have emerged as attractive targets for PDAC therapy (Table 1).

TGF-β, produced by CAF and being their main activator, enhances tumor progression and dampens the immune and therapy responses [126,127]. Thus, it represents a promising target to interfere with CAF function [128]. However, an early phase clinical trial testing the therapeutic efficacy of pharmacological TGF-blockade with the small molecule inhibitor Galunisertib in combination with chemotherapy for the treatment of unresectable PDAC did not reveal convincing results [129]. Additionally, data from a phase Ib clinical trial investigating Galunisertib in combination with the anti-PD-L1 antibody Durvalumab did not yield clinical improvements for patients with metastatic PDAC [130]. Of note, Özdemir et al. showed in a PDAC mouse model that complete depletion of α-SMA+ myofibroblasts led to enhanced tumor growth, decreased immune cell infiltration and increased number of Treg in the TME. These findings indicate that complete depletion of CAF is not a reasonable treatment strategy, as tumor-restraining CAF populations are eliminated, too. However, concomitant depletion of CAF and blocking of CTLA-4 decreased tumor growth and prolonged survival of PDAC bearing mice [131].

Furthermore, it has been shown that tumor derived ligands of the Sonic Hedgehog (Shh) signaling system leads to activation of stellate cells into CAF [132]. Based on this finding, Shh inhibitors were tested in preclinical and clinical trials to improve PDAC treatment. However, depletion or pharmacological inhibition of Shh reduced stroma formation but accelerated tumor growth of tumor bearing KPC mice [133]. In line with this finding, combined treatment of metastatic PDAC patients with Gemcitabine and Shh inhibitors, targeting the signal transducing component Smoothened (SMO), showed no superior effect compared to Gemcitabine monotherapy [134]. Another clinical trial investigated the efficacy of the SMO inhibitor Saridegib (IPI-926) in combination with FOLFIRINOX for treatment of patients with advanced PDAC demonstrating an antitumor activity along with acceptable safety [135]. However, since a study demonstrated that patients being treated with IPI-926 in combination with Gemcitabine showed even shorter survival times than patients treated with Gemcitabine alone, this study was terminated prematurely (see Infinity Pharmaceuticals company website) and other trials investigating the therapeutic efficacy of Hh inhibitors were withdrawn or stopped.

Due to the tumor boosting effect, achieved by depleting the entire CAF population, recent research has rather focused on stromal reprogramming than on stroma depletion by e.g., reversal of CAF activation or shifting the CAF populations. Since activated myofibroblasts can be reversed into a quiescent phenotype by retinoic acid [136], all-trans retinoic acid (ATRA) has been used for myofibroblast reversal. Thus, ATRA treatment of tumor bearing KPC mice resulted in an enhanced CTL infiltration in the tumor tissue, which was explained by reduced secretion of the chemokine CXCL12 by reverted stellate cells [88]. Another strategy aiming at a stromal (CAF) reprogramming is the treatment with Calcipotriol, which is a derivative of vitamin D. The vitamin D receptor itself is a transcriptional factor suppressing the activation of PSC. Treatment with Calcipotriol led to stroma remodeling and combined treatment with Gemcitabine reduced tumor growth and increased overall survival of PDAC bearing mice [137]. Recently, Biffi et al. demonstrated a phenotypic shift from tumor-promoting iCAF to tumor suppressive myCAF by JAK inhibition, which was associated with reduced tumor growth and increased collagen deposition in tumor bearing KPC mice, further impairing drug penetration into the tumor stroma [70]. This was also seen in clinical trials where the combination of a JAK inhibitor and chemotherapy did not prolong the overall survival [138].

**Table 1 cancers-13-04932-t001:** Overview of the cited therapeutic approaches targeting carcinoma associated fibroblasts (CAF).

Study System	Targeting Strategy	Reference
**Preclinical studies**		
PKT mouse model	Myofibroblast depletion by Ganciclovir	[131]
PKT mouse model	Myofibroblast depletion by Ganciclovir + anti-CTLA-4 antibody	[131]
KPC mouse model	Depletion or pharmacological inhibition of Shh	[133]
KPC mouse model	Myofibroblast reversal by ATRA	[88]
KPC mouse model	Vitamin D derivative Calcipotriol + Gemcitabine	[137]
KPC mouse model	Phenotype shift by JAK inhibitor (AZD1480)	[70]
**Clinical trials**		
Phase 1b/2 clinical trial	TGF-β blockade by Galusertinib + Gemcitabine	[129]
Phase 1b clinical trial	TGF-β blockade by Galusertinib + PD-L1 antibody Durvalumab	[130]
Phase 1b/2 clinical trial	Shh blockade by Vismodegib + Gemcitabine	[134]
Phase 1 clinical trial	SMO inhibition by Saridegib (IPI-926) + FOLFIRINOX	[135]
Phase 3 clinical trial	JAK1/JAK2 inhibition by Ruxolitinib + Capecitabine	[138]

Taken together, CAF represent a heterogeneous cell population in the TME of PDAC at the primary site as well as in metastases exerting diverse functions on other stromal cells (immune cells, EC) and PDAC cells. Importantly, CAF and their quiescent/non-activated counterpart, the stellate cells may exert tumor-restraining functions, which have been taken into consideration for the development of stroma-targeted therapeutic strategies to improve PDAC treatment.

### 3.3. Heterogeneity of Macrophages

Besides CAF, macrophages are also found in high numbers in many tumor entities representing the most abundant leucocyte population in the TME [139], also applying to PDAC [140]. Other, but less abundant, myeloid cell populations in PDAC are DC and MDSC. Tjomsland et al. showed that increased numbers of DC in the peripheral blood are associated with a better survival in PDAC patients [141]. However, most human PDAC specimens are characterized by low numbers of DC [142] and even if DC are present, they are mostly located at the tumor margins [142], a finding which could be similarly found in a modified KPC mouse model [143]. DC comprise a variety of subtypes including conventional and plasmacytoid DC exhibiting distinct phenotypes [144]. Of note, DC are key APC for the activation of T cells. Accordingly, these cells essentially determine the efficacy of anti-tumor immunosurveillance and immunotherapeutic treatment strategies, such as immune checkpoint inhibition and vaccinations. As described for other stromal cell populations, also MDSC can exhibit different phenotypes. In humans, three major MDSC subsets were identified: polymorphonuclear (PMN)-MDSCs, monocytic (M)-MDSCs and “early-stage MDSCs” (e-MDSC) [145]. Trovato et al. demonstrated that the MDSC frequency in PDAC was significantly correlated with overall survival and metastatic disease in PDAC patients, but only in some patients the immunosuppressive activity of purified MSDC was detectable. However, this finding was ascribed only to the monocytic subset [146]. One reason for the activity and expansion of MDSC in PDAC seems to be CD200, a regulator of myeloid cell activity, which is expressed in the PDAC TME. Choueiry et al. showed that MDSC isolated from human PDAC tissues express elevated levels of the CD200 receptor, by which CD200 can activate MDSC. Additionally, they found in a mouse model that the CD200 blockade impaired tumor progression and enhanced the efficacy of PD-1 immune checkpoint inhibition [147] supporting the immunosuppressive nature of these cells. Besides, it was shown that MDSC decrease T cell proliferation and lead to an enhanced apoptosis of activated T cells [148]. However, the exact mechanisms by which MDSC exert their immunosuppressive functions are still not fully elucidated and we refer to other excellent reviews on this cell type [149].

Owing to their prominent abundance in PDAC, we will focus in the following text mainly on the recent knowledge on TAM. TAM mainly originate from tissue resident macrophages or from blood derived monocytes which infiltrate into the injured tissue (e.g., due to cell infection or otherwise induced cell damage). Ontogeny studies in a murine PDAC model revealed that monocyte-derived TAM acquire functions such as antigen presentation, whereas embryonically-derived TAM show a more pro-fibrotic transcriptional profile indicating their role in ECM production and relating the origin of TAM to distinct functional phenotypes [150]. Importantly, in dependence on the environmental factors, monocytes/macrophages acquire different phenotypes exerting distinct functional effects. According to a simplified model, macrophages can differentiate into M1- and M2-macrophages. While interferon-gamma (IFN-γ) and bacterial lipopolysaccharide (LPS) promote polarization of pro-inflammatory M1-macrophages, IL-4, IL-13, IL-10 and TGF-β foster an anti-inflammatory M2-phenotype in humans [151,152,153]. Thus, the composition of the TME (e.g., oxygen level, amount and type of other stromal or tumor cells) impacts phenotype and effector function of macrophages and promotes the switch from one type to another, a process which also occurs under physiological conditions [154]. As outlined above, CAF promote an immunosuppressive environment e.g., via CXCL12 and M-CSF secretion thereby recruiting monocytes and promoting the accumulation of M2-polarized macrophages [90,155,156]. Moreover, a hypoxic TME provokes an M2-polarization, while an oxygen enriched microenvironment drives M1-polarization [83]. In a murine PDAC model, PDAC cell derived small extracellular vesicles (sEV) covered with Ezrin were shown to foster a polarization towards an M2-phenotype and supporting metastasis formation in the liver [157]. Studies in another PDAC mouse model showed that uptake of PDAC-derived exosomes by Kupffer cells contributes to formation of a pre-metastatic niche in the liver involving the recruitment of bone marrow-derived macrophages [105]. Granulin, expressed by circulating monocytes and hepatic metastasis-associated macrophages, leads to activation of HSC into HMF thereby promoting liver metastasis formation [158]. Thus, CAF and TAM maintain a vicious stimulatory cycle that sustains and further fuels itself, contributing to a tumor-promoting TME.

M1-macrophages secrete different pro-inflammatory cytokines such as IL-12, IL-23 and express elevated levels of B7 family members (B7-1, B7-2) and MHC II for appropriate activation of a TH1 response by which these cells foster anti-tumor effects [159]. In contrast, M2-macrophages secrete anti-inflammatory cytokines such as IL-10, TGF-β and others to prevent T cells from effectively exerting their anti-tumor functions [160].

Although the frequency and phenotype of macrophages are subject to high dynamic variation of the pancreatic microenvironment during tumorigenesis [140], TAM in PDAC predominantly exhibit M2-characteristics. These are associated with diverse pro-tumoral effects including promotion of tumorigenesis, immunosuppression, metastasis acceleration and chemotherapeutic resistance [159]. In primary PDAC, a high infiltration of macrophages, particularly CD163+ macrophages, is observed [24,140,161,162,163], indicating an M2-type which has been associated with a poor prognosis and decreased survival [162,163,164,165]. Additionally, the number of TAM is inversely associated with the maturity of the tumor stroma, as an immature (highly cellular and collagen poor) stroma correlates with an increased the number of TAM [163]. As M2-macrophages secrete a plethora of growth factors, cytokines and chemokines (e.g., IL-10, IL-6, Tumor Necrosis Factor-alpha (TNF-α), TGF-β, chemokine (C-C motif) ligand 20 (CCL20)), they are potent inducers of the EMT [161,166]. Furthermore, PDAC tumor growth and metastasis formation are dependent on phosphatidylinositol 3-kinases (PI3Kγ) expression in macrophages [167]. Of note, TAM can co-exist as M1- or M2-macrophages or also intermediate forms in the TME which are associated with anti- or pro-tumoral functions [168]. Accordingly, Helm et al. could demonstrate that human PDAC-derived TAM can exhibit both M1- and M2-properties and that also M1-macrophages are able to increase migratory and invasive abilities of PDAC cells [161,168]. In line with these findings, Chen et al. showed that M2-polarized macrophages cause inflammation via IL-1β release. This increased IL-1β release in turn promotes EMT induction and thereby formation of metastases in an orthotropic PDAC mouse model [169].

Macrophages also contribute to apoptosis and therapy resistance of PDAC. Buchholz et al. showed that murine and human macrophages can rapidly metabolize and inactivate the chemotherapeutic drug Gemcitabine resulting in profound therapy resistance [15]. Another study could show that murine bone marrow derived macrophages when exposed to PDAC cells release pyrimidine species, among others deoxycytidine, which inhibits gemcitabine through molecular competition representing another mechanism of macrophage mediated chemoresistance [170]. Liu et al. suggest that macrophages contribute to immunosuppression and failure of immunotherapies by demonstrating an accumulation of MDSC and M2-polarized TAM in tumoral lesions of different PDAC mouse models which is accompanied by a reduction of CTL and T helper cells. Besides, the number of M2-macrophages increased after application of Gemcitabine in this model system, which could be another mechanism for drug resistance [171]. Weizman et al. identified in humans another TAM-mediated mechanism of chemoresistance via upregulation of cytidine deaminase (CDA) which leads to reduction of Gemcitabine-induced apoptosis. CDA is the enzyme which transports chemotherapeutics into cells and is responsible for their metabolization [172]. Using a genetic mouse model, Binenbaum et al. revealed that TAM communicate with cancer cells via so called macrophage-derived exosomes which are internalized by cancer cells leading to reduced Gemcitabine concentrations by upregulation of CDA in the PDAC cells [173]. Moreover, tumor-derived exosomes can be taken up by human and murine macrophages, thereby increasing expression of the immune checkpoint molecule PD-L1 on these cells and promoting macrophage-mediated immune suppression. Uptake of tumor-derived exosomes also alters the cytokine secretion (IL-6, IL-1β, IL-10 and TNFα) of macrophages thereby facilitating several pro-tumoral functions [174].

### 3.4. Approaches for Targeted Therapy of Macrophages

As macrophages exert diverse pro-tumoral effects e.g., promoting an immunosuppressive TME and drug resistance, these stromal cells have been qualified as a reasonable target for cancer therapy (Table 2). Accordingly, several preclinical and clinical trials targeting macrophages have been conducted. One approach to simultaneously harness multiple immune cell types including macrophages against cancer cells is the treatment with CD40 agonists [175]. CD40 is a cell surface receptor primarily expressed by B cells, DC and myeloid cells including macrophages. Through interaction with its ligand CD40L (expressed on CD4+ T cells), CD40 is activated. This activation results in various physiological effects including increased expression of MHC II and T cell costimulatory receptors on its target cells [175,176]. Consequently, antigen presentation and CTL activation are promoted. Moreover, in the KPC mouse model, CD40 agonist treatment resulted in macrophage activation with increased expression of matrix metalloproteinases. Via this mechanism, macrophages reduced the pronounced PDAC-associated desmoplasia [177]. Recently, macrophage-mediated stroma depletion and reprogramming towards an M2-phenotype were confirmed for human PDAC by translational data obtained from a clinical trial that assessed the CD40 agonist Selicrelumab [178]. Moreover, in a PDAC mouse model, CD40 agonists synergized with checkpoint inhibitors [179]. Whilst early clinical trials suggested a positive clinical effect of CD40 agonist monotherapy or combination with Nivolumab [177,180], this could not be confirmed in phase II clinical trials [181]. However, multiple clinical trials evaluating CD40 agonists for the treatment of PDAC in combination with chemotherapeutic or targeted agents are ongoing (NCT03214250; NCT04536077; NCT04807972; NCT04888312) and may yield more promising results. Furthermore, the combination of an anti-PD-1 antibody with Gemcitabine showed a beneficial effect in a murine PDAC liver metastasis model, which was explained by an increase of M1-macrophages and a TH1 response [182].

Since specific targeting of macrophages spares other immunosuppressive monocytic cell populations which may impair the therapeutic efficacy, targeting of the integrin CD11b/CD18 which highly expressed on several myeloid cell subsets has been suggested as a promising strategy for PDAC treatment. Using the CD11b small-molecule agonist (ADH-503) in KPC mice, a partial activation of CD11b along with TAM repolarization was observed. In addition, increased numbers of immunosuppressive myeloid cells in tumor tissues were observed as well as enhanced dendritic cell activity. Thus, this strategy targeting multiple immunosuppressive myeloid cells led to an improved antitumor T cell response which might be even more effective in combination with immune checkpoint inhibitors [183].

However, to date no convincing strategies targeting macrophages have been elaborated and implemented in treatment of PDAC patients. One explanation might be that as demonstrated for CAF, macrophages represent a heterogeneous cell population exerting highly context-dependent pro- and anti-tumoral effects. Thus, this dynamic phenotype switching, which also seemed to be influenced by the different therapeutic strategies, have to be considered in an effective and sustainable anti-tumor therapy of PDAC patients.

### 3.5. Heterogeneity of T Cells

Although we focus on T lymphocytes in this article, we want to briefly mention the other group of lymphocytes, namely the B cells. The role of B cells in PDAC is still controversially discussed but in general, B cells are described to be rather tumor promoting. Thus, Pylayeva-Gupta et al. showed that B cells secrete IL-35, which promotes the proliferation of tumor cells [184]. Furthermore, coculture of B cells isolated from PDAC patients enhanced the production of collagen by fibroblasts, resulting in ECM remodeling [185]. In a mouse model, it was shown that B cells are recruited by the B cell chemoattractant CXCL13 already during PanIN formation. Similarly, accumulation of B cells was also observed in PanIN lesions in PDAC patients [184,185]. Overall, there are not a lot of studies focusing on B cells in PDAC but it might be useful to investigate further on B cells in order to understand the complex microenvironment better.

Besides diverse CAF and macrophage phenotypes, different T cell populations can be found in the TME of the primary tumor and metastases in PDAC patients. Owing to their different effector phenotypes, T cells essentially impact the process of tumor and metastases formation in several ways. In general, an increase of immunosuppressive (T) cells can be observed during PDAC development, while tumor directed immune functions are impaired and/or even lost [140,186,187]. Key players of the immune response against tumor cells are CD8+ CTL. Accordingly, several studies showed that a high tumor infiltration of CD8+ CTL is associated with a longer overall survival of PDAC patients [188,189,190,191].

Rahn et al. showed in human PDAC tissues that CTL are predominantly present in the TME and less in close proximity to PDAC cells [24]. One reason for this can be seen in the desmoplastic stroma containing the ECM which acts as a physical barrier for immune cells and in particular CTL [69,192]. Similar findings were reported on liver metastases. Accordingly, in a KPC mouse model small metastatic lesions in the liver show a high CD8+ T cell infiltration compared to large metastatic lesions. Furthermore, in small lesions CD8+ T cells express CD69 and no PD-1, indicating that these CTL still exhibit their effector phenotype, while in large metastatic lesions CTL express PD-1 but no CD69, indicating T cell exhaustion [193]. Importantly, this exhausted state is characterized by elevated expression of inhibitory receptors (PD-1, CTLA-4 and T cell immunoglobulin domain and mucin domain-3 (TIM-3)), decreased production of cytotoxic molecules (perforin, granzyme A/B and granulysin), decreased production of chemokines (TNF-α and IFN-γ) and higher CTL apoptosis [194,195]. For optimal CTL survival and effector function, CD4+ T helper cells are pivotal. In line with this finding, a high tumor infiltration of CD8+ CTL together with CD4+ T cells correlate with a better prognosis for PDAC patients [188,189]. CD4+ T cells are another very heterogeneous T cell population, as they can differentiate into divergent subsets (TH1, TH2, TH17 and Treg). The differentiation of TH0 helper cells into different subsets is dependent on cytokines (IL-12, IL-4, TGFβ, IL-6 and IL-2) in the microenvironment [196]. The different subsets are characterized by expression of distinct cytokines and therefore show divergent impacts on all cells in the TME. Similar to CAF and macrophages, the effects can be either tumor-promoting or tumor-suppressing.

TH1 cells are regarded as tumor-suppressing as they exert various immune response activating functions e.g., they release IFNγ, which promotes recruitment of CTL, M1-macrophages and Natural Killer (NK) cells, and IL-2 which activates CTL [197,198]. On the other hand, IFNγ also induces PD-L1 expression on tumor cells, T cells, myofibroblasts and macrophages, thereby supporting CTL inhibition and immune escape in PDAC [199].

However, PDAC cells and the TME rather promote differentiation of TH2 cells which are regarded as immune-suppressing and thereby tumor-promoting [200], mainly because of the release of cytokines like IL-4, IL-5, IL-6, IL-10 and IL-13 [187]. IL-4 and IL-10 foster the differentiation of monocytes into M2-macrophages [201], IL-4 and IL-13 trigger the collagen synthesis in myofibroblasts thereby contributing to ECM remodeling [202] and IL-13 has been shown to enhance growth of PDAC cells [203]. Accordingly, elevated levels of TH2 cytokines can be detected in plasma samples [187] and the TME of PDAC patients comprises higher numbers of TH2 cells compared to TH1 cells [204]. Moreover, high plasma levels of TH2 cytokines in patients with resectable PDAC are associated with a shorter survival [205]. 

Furthermore, human PDAC tissues contain higher numbers of TH17 cells compared to normal pancreatic tissue, which is associated with a shorter median survival of PDAC patients [206]. TH17 cells are regarded mainly as tumor-promoting because this T cell subset is characterized by elevated release of IL-17, IL-21 and IL-22 [197]. Importantly, IL-17 has been shown to enhance initiation and progression of PanIN in a murine PanIN model, thus being a trigger in early pancreatic tumorigenesis [207].

Another important immunosuppressive cell population in PDAC tissues are Treg. Especially in human PDAC, Tregs are mostly located in the stroma and only rarely in the epithelial layer of pancreatic ducts. Furthermore, Hiraoka et al. demonstrated a significant increase in the number of Treg during progression from low grade PanIN to an invasive PDAC. Additionally, a high prevalence of Treg in PDAC is significantly correlated with distant metastases, advanced tumor stage and high tumor grade as well as poorer prognosis [208,209]. The mechanisms by which Treg enrich in human and murine PDAC can be diverse e.g., tumor associated Treg can derive from peripheral Treg recruitment, expansion of tissue resident Treg, differentiation from local naïve T cells or conversion of conventional T cells [210]. Here, tumor- or CAF derived TGF-β can induce conversion of CD4+ CD25- T cells into Treg [211,212]. It was also shown that Forkhead box protein 3 (FoxP3) expressing tumor cells recruit Treg by directly trans-activating CCL5 [213]. Yang et al. observed a correlation of highly expressed secreted frizzled-related protein 4 (SFRP4) and Treg infiltration in tumors of KPC mice and PDAC patients [214]. Shen et al. speculated that PDAC cell-derived sEVs induce an enrichment of human FoxP3+ Treg and further an overexpression of immune checkpoint molecules PD-1, PD-L1, CTLA-4 and TIM-3 as well as an enrichment of FoxP3+ Treg [215]. Finally, L1 cell adhesion molecule (L1CAM) expression in the pancreatic ductal epithelium was shown to promote enrichment of human Treg in PDAC by e.g., enhancing migration of Treg, decreasing proliferation of CD4+ effector T cells and promoting conversion into a CD4+CD25-CD69+ regulatory T cell phenotype [216]. Moreover, these CD4+CD25-CD69+ Treg are detectable at high numbers in human PDAC tissues and correlate with nodal invasion and higher grading in PDAC patients [216]. One mechanism by which Treg suppress the anti-tumor response is by interaction of CTLA-4 with CD80/CD86 on DC leading to a reduced expression of MHC class II, CD40 and CD86 molecules, all being important for Treg maturation and activation. Besides, tumor-associated DC express Indolamine-pyrrole 2,3-dioxygenase (IDO), which suppress T cell responses and promote immune tolerance [217]. Moreover, Treg crosstalk with and are dependent on MDSCs as in a PDAC mouse model, depletion of MDSC led to a reduced recruitment and/or induction of Treg in pancreatic tumors and development/expansion of Treg seems to require a direct cell–cell interaction with MDSC [148]. Furthermore, TGF-β is not only a trigger for Treg development but also production of reactive oxygen species (ROS) by these cells contributing to oxidative stress in the TME [218].

Finally, γδ T cells represent a promising tumor-suppressive T cell population because of their ability to recognize antigens in an MHC-independent manner, to present antigens to CD3+ αβ T cells as well as their phagocytic properties. γδ T cells have a prevalence of 1–10% in the blood [219]. In human PDAC tissues, γδ T cells are mainly found in the tumor stroma or adjacent to or within ductal epithelium [220]. γδ T cell infiltration can be promoted by CAF derived CXCL12 [221,222]. However, the presence of γδ T cells is not sufficient to exert potent anti-tumor responses, because it has been shown that Galectin-3 which is expressed on PDAC cells but also αβ- and γδ T cells, inhibits T cell proliferation and is thus regarded as an intrinsic tumor escape mechanism [223,224]. Furthermore, kynurenine, a downstream metabolite of IDO, was identified as an inhibitor of γδ T cell cytotoxicity and proliferation in PDAC [225]. However, Daley et al. showed that γδ T cells can inhibit αβ T cell activation via checkpoint receptor ligation leading to suppression of CD4+ and CD8+ T cells and acquisition of an activated CD44+CD62L- phenotype. Additionally, γδ T cells were shown to decrease TNF-α expression of αβ T cells in vitro. Importantly, these processes can be reversed by PD-L1 blockade suggesting that γδ T cells are important modulators of a checkpoint receptor-dependent immunosuppression and pinpoint also to a tumor-suppressing role of these cells in PDAC [226].

First attempts have been undertaken to consider this T cell heterogeneity to provide a rationale for patient stratification and optimized treatment choice. Depending on the number of infiltrating lymphocytes, tumors have been categorized into immunological subtypes: T cell inflamed (“hot tumors”) or non-T cell inflamed (“cold tumors”). Owing to the low T cell infiltration especially of CD8+ CTL, PDAC is mostly characterized as a cold tumor [227]. Importantly, immunotherapies are often not effective in “cold” tumors explaining their common failure in PDAC [25,228].

### 3.6. Approaches for Targeted Therapy of T Cells

Since single agent immunotherapy with cancer vaccines or immune checkpoint inhibitors has failed so far in clinical trials [21,22,229,230], multi-agent combinations and combinations with radiotherapy have been evaluated as treatment option for PDAC. In order to boost T cell responses against cancer cells, combinations of two checkpoint inhibitors (Durvalumab and Tremelimumab) targeting the CTLA-4 and PD-1/PDL-1 axes have been evaluated [231,232]. Moreover, clinical trials were designed to assess combinations of cancer vaccines (e.g., GVAX and CRS-207) and checkpoint inhibitors or checkpoint inhibitors with oncolytic viruses (Pelareorep) or chemokine receptor inhibitors (BL-8040) based on promising preclinical data [233,234,235,236]. To date these approaches have not yielded relevant clinical improvements for PDAC patients. However, a high number of clinical trials investigating innovative immune checkpoint inhibitor-based combinations (e.g., with IL-6 antagonists (NCT04258150), agonists of stimulator of interferon genes (NCT03010176) and kinase inhibitors (NCT04820179) are underway.

As outlined above, γδ T cells represent an attractive effector T cell population for cancer therapy as they recognize antigens and kill target cells MHC independently. Thus, bispecific Abs binding CD3 or Vγ9 on γδ T cells and HER2/neu on PDAC cells enhanced the cytotoxicity of γδ T cells via granzyme B and perforin release and led to a reduced tumor growth in a subcutaneous PDAC Xenograft model [219]. Furthermore, the tribody [(HER2)2xCD16] activated human γδ T cells and NK cells to lyse HER2 expressing PDAC cells via granzyme B release [237]. Additionally, monitoring of γδ T cells subpopulations in PDAC patient’s blood and determination of their cytotoxicity can also help to optimize γδ T cell-based immunotherapy [238].

Depleting Treg could also be an option to improve PDAC therapy. As Treg express CCR5, a current phase1/2 clinical trial investigates the therapeutic efficacy of CCR2 and CCR5 inhibitors in advanced PDAC with the aim to reduce Treg, MDSC and M2-polarized TAM and to increase anti-tumor immunity [209]. The bispecific antibody ATOR-1015 [CTLA-4xOX40] induced CD8+ T cell activation and Treg depletion in several syngeneic tumor models including a PDAC model, thereby resulting in a tumor-specific and long-term immunological memory. Furthermore, ATOR-1015 enhanced the response to PD-1 blockade, thus providing a reasonable combination with PD-L1 inhibitors [239]. However, depletion of the entire Treg population might bear the risk for the development of autoimmunity. Additionally, preclinical studies in a PDAC mouse model demonstrated that Treg depletion does not lead to diminished immunosuppressive activity and accelerates tumor growth. Since Tregs release high amounts TGF-β, Treg depletion resulted in a reduced TGF-β levels by which in turn tumor-restraining α-SMA+ CAF were reprogramed into iCAFs. Furthermore, increased numbers of myeloid cells were observed in the tumors indicating that specific targeting of this particular T cell population essentially alters the stroma cell dynamics in the TME by which unexpected tumor promoting effects might be induced [240].

In summary, PDAC-associated T cells are characterized by a great diversity exemplified by their abundance (high versus low numbers), spatial localization (in close proximity to tumor cells versus located in the TME) and effector phenotype (activated versus anergy/exhausted). Adding to this huge complexity, distinct T cell populations exert both tumor-promoting and tumor-suppressing functions. Importantly, the immunological TME composition and overall immunity in a tumor and patient, respectively, seem to be highly dependent on a variety of host intrinsic (e.g., genetics) and extrinsic factors (e.g., exposure to pathogens or environmental factors) [241] defining the above-mentioned parameters which have to be considered for an effective anti-PDAC therapy.

### 3.7. Heterogeneity of Endothelial Cells

In general, human PDAC tissues exhibit a high microvascular density and poorly perfused blood vessels, the latter showing a heterogeneous distribution pattern in PDAC subtypes [242,243]. Moreover, a high microvascular density along with a diminished integrity of these microvessels is associated with disease progression and poorer survival in PDAC patients [244]. This immature vasculature and the high intratumoral pressure in PDAC due to the pronounced desmoplastic stroma results in an irregular as well as disorganized tubular architecture. Murine and human studies revealed that this in turn impairs drug delivery, suppresses lymphocyte infiltration and increases hypoxia within the tumor mass [133,245,246,247]. In general, blood vessels support tumor growth by enabling influx of nutrients and oxygen. In addition, EC also affect tumor cells, stromal cells in the TME and therapy responses [248,249,250].

Tumor angiogenesis is induced by proangiogenic factors such as VEGF released by tumor and stromal cells [251]. Upon this “endothelial cell activation”, changes in the gene expression profile and phenotypes of EC occur by which these cells exert their pleiotropic effects in the tumor [252,253]. Thus, increased expression levels of Vascular Endothelial Growth Factor Receptor (VEGF-R), EGF-R and diverse cell adhesion molecules (e.g., Intercellular Adhesion Molecule (ICAM), Vascular Cell Adhesion Molecule (VCAM), E-selectin) as well as an increased secretion of cytokines (e.g., IL-8) are observed in activated EC or tumor associated endothelial cells (TEC). These alterations essentially contribute to tumor inflammation and immune evasion in human as well as in murine tumors [254,255,256,257,258,259,260]. Thus, human derived material as well as murine xenografts revealed that TEC are not only morphologically distinct from their physiological counterparts, but also exhibit diverse genetic and phenotypic alterations by which these cells impact tumor development and progression as well as therapy responses. Furthermore, TEC can undergo Endothelial-Mesenchymal-Transition thereby giving rise to CAF and contributing to CAF enrichment in PDAC. In humans, this process seems to be mediated by inflammatory factors such as TNF-α [261,262,263].

Furthermore, Issa et al. demonstrated that upregulation of L1CAM expression on human PDAC-derived TEC increase adhesion and transmigration of PDAC cells to and through the PDAC-derived endothelium, thereby facilitating PDAC metastasis [264]. Sano et al. described that elevated expression of VCAM-1 on TEC promotes progression of PDAC as well as PanIN in a genetically engineered mouse model, but also increases the incidence of cancer-associated thrombosis in mice and patients, which is known as a poor prognostic factor in PDAC. Moreover, this is accompanied by an increased infiltration of immunosuppressive leucocytes into the tumor tissues. Blocking VCAM-1 reduces the number of thrombotic/thromboembolic events as well as infiltration of TAM and tumor-associated neutrophils and prolonged overall survival of treated mice [265]. In line with these findings, PDAC-derived TEC exhibit an enhanced expression of distinct adhesion molecules (e.g., E-selectin, ICAM-1, VCAM-1, Mucosal vascular address in cell adhesion molecule 1 (MAdCAM-1)) compared to EC from healthy pancreas. These PDAC-derived EC promote transendothelial migration of Treg thereby contributing to the immunosuppressive TME in human as well as murine PDAC [266]. In contrast, downregulation of these adhesion molecules (e.g., ICAM-1/2, VCAM-1, MAdCAM-1, E-selectin) on TEC leads to a decreased infiltration of putative tumor-directed lymphocytes into the TME [209,267].

### 3.8. Approaches for Targeted Therapy of Endothelial Cells/Angiogenesis

Overall, these data indicate that TEC can exhibit different phenotypes and functions compared to their physiological counterpart, which is dependent on the tumor entity and TME composition e.g., the amount of macrophages which are a main source of VEGF. Importantly, the aberrant morphology of TEC and their altered expression profile of adhesion molecules are essential determinants for various malignancy-associated processes, such as immune evasion and drug resistance in PDAC patients [268]. Accordingly, TEC and angiogenesis, the latter being a hallmark of cancer, have been identified as promising targets for cancer therapy and many anti-angiogenic strategies have meanwhile been approved for the treatment of a variety of cancer entities. Demonstrated in preclinical and clinical studies, those strategies primarily aim at the inhibition of angiogenesis, thereby reducing nutrient supply and deactivating EC on the one hand [269,270,271,272] and normalizing the vasculature to improve drug delivery and immune cell access into the TME on the other hand [273,274,275,276]. Two commonly used anti-angiogenic drugs are Bevacizumab and Aflibercept, both binding to VEGF, and Aflibercept additionally binding to Placenta-derived Growth Factor (PGF). In addition, tyrosine kinase inhibitors (such as Sunitinib, Sorafenib, Pazopanib) that target VEGF-R2 e.g., expressed by TEC [277] are also FDA approved for anti-angiogenic cancer therapy. However, the above-mentioned strategies still fail to significantly improve overall or progression free survival of PDAC patients [278,279].

Accordingly, intensive research efforts are ongoing to identify predictive markers that allow for rational patient stratification in order to select the optimal therapeutic strategy to specifically target the aberrant vasculature in PDAC [280]. Considering the pleiotropic effects of TEC in PDAC, TEC targeting may be more effective in combination with e.g., immune checkpoint inhibitors or modified drugs (e.g., encapsulated forms) to optimize treatment strategies. Furthermore, more studies are required investigating the optimal timing for anti-angiogenetic therapies (particularly in combination with other therapies) in order to exert the maximum effect.

### 3.9. Heterogeneity of the Microbiome

In recent years compelling evidence has emerged that not only the TME but also the microbiome is altered in PDAC patients—in the tumor as well as in other body compartments. Experimental data support the view that the altered microbiome is not just a surrogate of the disease but rather essentially impacts development, progression and therapy responses of PDAC [11,19,20]. Miller et al. also identified an influence of the microbiome on the TME, thereby promoting PDAC development and explaining reduced therapeutic efficacy of several agents. Hence, ablation of the microbiome was shown to prevent PDAC in preclinical studies [281].

It is still not fully understood how bacteria, detectable in PDAC tissues, enter the pancreas. Potential mechanisms include the oral route or translocation from the lower gastrointestinal tract through the portal circulation or mesenteric lymph nodes [282]. Thus, it has been shown that human cystic pancreatic precursor lesions contain Fusobacterium nucleatum and Granulicatella adiacens, both bacterial populations commonly found in the oral cavity [283,284]. In line with this finding, increased amounts of Fusobacterium species were detected in PDAC tissues and correlated with a worse prognosis of PDAC patients [285]. Another study demonstrated that Gammaproteobacteria are the predominant bacterial species in human PDAC tissues, most of them being members of Enterobacteriaceae and Pseudomonadacea families [19].

It has been shown in patient suffering from malignant melanoma and lung cancer that the gut microbiome impacts the response to immunotherapy and chemotherapy. This finding suggests that the composition of the gut microbiome impacts activation of the immune system [286,287,288] and thereby promotes cancer-associated inflammation, a mechanism which might also be relevant in PDAC. Of note, chronic pancreatitis (CP), being a risk factor for PDAC development, can be caused by microbial infections [289].

As outlined above, an inflammatory and desmoplastic environment supports various malignancy-associated processes, such as immune evasion and metastasis and impacts therapy responses [290,291,292], all processes related to EMT [293]. Importantly, a tumor-associated microbiome was shown to promote EMT, thereby leading to epithelial barrier alterations and tumor-promoting inflammation. In detail, it could be shown that infections by Fusebacterium nucleatum lead to the loss of membranous E-cadherin [294], a key step in EMT [11,295,296,297]. Owing to the fact that high levels of Fusebacterium are found in human pancreatic precursor lesions and PDAC tissues, it is reasonable that this might be one mechanism by which early EMT is induced in pancreatic tissues. Of note, EMT-associated alterations are already found in PDEC in human CP tissues [296] and lead to early PDEC dissemination prior to tumor formation in the pancreas in a murine mouse model [297]. Zhang et al. postulated that each microbial pathogen conquering the pancreatic tissue has the potential to induce EMT-related pathological changes [11]. Since EMT induction has been also linked to the acquisition of CSC properties and drug resistance [2,6,10], EMT/CSC induction might represent another mechanism by which the altered microbiome contributes to the pronounced therapy resistance in PDAC.

In this context, Geller et al. reported not only the presence of Gammaproteobacteria in human PDAC tissues but also that these bacteria, expressing the long form of the enzyme CDA, are able to metabolize the cytostatic drug Gemcitabine into its inactive form. These findings suggest that the presence of Gammaproteobacteria in PDAC tissues may contribute to Gemcitabine resistance in these patients [19].

Moreover, Riquelme et al. discriminated long-term survivor (LTS) PDAC patients showing a high bacterial alpha diversity from short-time survivor (STS) PDAC patients with a lower alpha diversity. They also demonstrated that the microbiome composition in PDAC tissues differs from that in normal tissues and that the gut microbiome accounts for 25% of the tumor microbiome, supporting translocation of bacteria from the gut into pancreatic tissues as one mechanism for bacterial PDAC tissue colonization [20]. Additionally, they determined greater numbers of CD3+ T cells, CD8+ T cells and granzyme B+ cells in LTS compared to STS PDAC patients, thus correlating a higher CTL infiltration with a higher microbiome diversity. These data indicate that the tumor microbiome composition influences both the extent of immune infiltration and the degree of CD8+ T cell activation in PDAC tissues [20]. It also suggests that alterations of the microbiome directly in the tumor but also in other body compartments (e.g., the gut) essentially contribute to shaping the patient’s immune response. In line with these findings, it was shown in different tumor mouse models that the efficacy of the CTLA-4 inhibitor Ipilimumab depends on the presence of distinct commensal gut *Bacteroides* species (spp.) which apparently trigger TH1 immune responses, being important for the anti-tumor effects. Hence, administration of antibiotics decreased the therapeutic effect of Ipilimumab [298]. Moreover, this study also demonstrated that fecal microbiota transplantation (FMT) of human *Bacteroides* spp.-rich feces significantly increases the overall outcome of tumor bearing animals by markedly increasing the response to CTLA-4 blockade. Supporting these data, Tanoue et al. detected an 11-strain consortium of bacteria isolated from gut microbiota of human healthy control donors (HC) which can induce IFN-g+ CD8+ T cells to enhance the efficacy of immune checkpoint inhibitors in tumor-bearing mice [299]. Underscoring the therapeutic potential of FMT, a significant reduction in tumor growth was observed in PDAC bearing mice after FMT from PDAC LTS donors with no evidence of disease (LTS-NED) compared to mice obtaining FMT from STS patients or HC [20]. Moreover, tumors from mice treated with FMT of STS PDAC patients were larger than those from mice who received FMT of HC. Finally, short-term antibiotic treatment of tumor bearing mice which had received FMT from LTS-NED donors led to even larger tumors than those of untreated mice. These data suggest that certain PDAC-associated bacteria exert a tumor-promoting effect on the one hand and that bacterial ablation by antibiotic treatment decreases the anti-tumoral efficacy of distinct bacterial species e.g., found LTS PDAC patients on the other hand.

### 3.10. Approaches for Microbiome Modulating Therapies

As already mentioned above, studies provide evidence that the response of cancer patients to blockade of the immune checkpoints PD-1/PD-L1 or CTLA-4 is influenced by the gut microbiome [298,299]. Tumor mouse models revealed that different microbiome compositions cause significant differences in response to treatment with PD-1/PD-L1 inhibitors [298] so that probiotics have been suggested as part of the anti-cancer therapy. Accordingly, Sivan et al. demonstrated reduced tumor growth and a beneficial response to anti-PD-1 therapy in melanoma bearing mice after oral administration of Bifidobacterium spp. containing probiotics. Furthermore, the superior anti-tumor effect from mice with a favorable microbiome composition could be transferred to other mice by FMT or co-housing [298,300]. In summary, these findings strongly support the therapeutic potential of FMT or certain diets/probiotics with the aim to restore the commensal (tumor-suppressing) microbiome and to displace the tumor-promoting microbiome. Combining such strategies with chemo- or immunotherapy might help to overcome therapy resistances and to improve PDAC patient prognosis.

## 4. Therapeutic Implications and Challenges of TME Targeting

The study by Rashid et al. revealed that consideration of different molecular subtypes e.g., those identified by [28,29,30], might be reasonable for stratifying PDAC patients for chemotherapy, as they provided evidence for efficacy prediction towards FOLFIRINOX treatment according to molecular tumor subtypes [32]. Adding to the multiple levels of tumor heterogeneity, stromal cell populations as well as the microbiome have been characterized by a high heterogeneity, too, implying both tumor-promoting and tumor-restraining functions. As outlined in the sections above, multiple therapeutic approaches have been developed and tested to target distinct TME components in order to improve treatment outcome of PDAC patients. However, many of these studies revealed unexpected results (e.g., by depleting the entire CAF population leading to even more aggressive tumors [131] or promising preclinical findings could not be confirmed in clinical trials [134]. Thus, until today, less than 5% PDAC patients receive a targeted therapy and significant improvements of survival are still missing [301]. Current clinical trials such as the “Precision-Panc” (NCT04161417) and the “Precision Promise” studies (NCT04229004) are ongoing with the aim to validate the therapeutic efficacy of novel individualized therapies in PDAC patients [302,303]. However, results from these trials are still pending.

The clinical failure of these strategies might also be explained by the use of inadequate preclinical models and/or an insufficient consideration of the stromal cell heterogeneity, to clearly discriminate which stromal phenotype is “friend or foe” [192,304,305]. For any kind of targeted and personalized therapy in cancer patients, pre-therapeutic samples and biopsies, respectively, are of pivotal importance and a critical determinant to assess inter- and intratumoral characteristics as well as changes during treatment courses. These allow identification of the most effective therapy and to eventually select therapies according to the profile identified [32]. In this context, surgically resected specimens provide the largest number of cells, reflect best tumor heterogeneity and enable pathologists to perform comprehensive analyses [306]. In cases of patients with an advanced disease, not being suitable for surgical resection or with recurrent tumors, fine needle aspirated (FNA) samples have been proven to be useful for identification of targets for targeted therapies. However, FNA samples only reflect a small portion of the tumor bearing the risk of not fully representing tumor and stroma heterogeneity [32,301,303]. Nonetheless, FNA-obtained samples are in certain cases the only option to obtain information on tumor and TME characteristics and help to provide individualized therapy to patients with metastatic disease whose tumor is not resectable. Liquid biopsies and analysis of cell free nucleic acids, circulating tumor cells or extracellular vesicles are additional methods to obtain information on tumor cell characteristics. Repeated blood draws (or collection of other body fluids) are more feasible than repeated biopsy sampling and may therefore complement tissue-based analysis for longitudinal monitoring of PDAC dynamics and therapy response [307]. However, important players of PDAC biology, i.e., the TME and the microbiome are not considered in these approaches.

Of note, Birnbaum et al. already suggested to consider not only the PDAC tumor subtypes but also the stromal subtypes classified by Collisson et al., Moffitt et al. and Bailey et al. for targeted therapy in PDAC patients [28,29,30,31]. Even though targeting the TME based on stromal subtypes might be a more efficient therapeutic approach, a challenge will be to fully consider differences in distribution and extent of the TME at different sites (primary tumor versus metastases) which might be associated with divergent functional effects on PDAC development. Furthermore, the TME might be further differentially modified by chemotherapy [303,308]. Additionally, Torphy et al. identified a high stroma content as a factor for a favorable outcome in patients suffering from metastatic PDAC [309], further underscoring that the TME heterogeneity is a critical determinant for targeted stromal therapy.

The current knowledge gained from comprehensive preclinical and clinical trials demonstrating a rather limited efficacy of tumor and TME targeted therapies in PDAC patients, leads us to the following assumptions: (i) due to its extraordinary complexity and peculiarity, the TME in PDAC easily compensates pathways, interactions and effects that are modulated by mono-targeted therapies resulting in limited therapeutic efficiency, (ii) targeted therapies applied so far have impaired both tumor-promoting and tumor-restraining TME populations, thereby counteracting the beneficial therapeutic effects albeit it was intended to solely target the tumor-promoting cell populations, (iii) the functional impact of certain host-derived factors on PDAC development and on shaping therapy responses are still insufficiently understood and considered (e.g., the host microbiome) and (iv) the usage of reliable biomarker (not only in tumor tissue but also other compartments) for therapy decision making has still to be improved for treatment of this tumor entity.

Accordingly, concepts aiming at restoration and normalization of a physiological, tumor-restraining microenvironment, respectively, rather than at precisely targeting distinct TME components might be more effective strategies [180,310]. For example, Kocher et al. focused on the stroma in patients with locally advanced or metastatic PDAC and replenished the Gemcitabine-Nab-Paclitaxel chemotherapy with ATRA in a phase I clinical trial. Clinical safe and tolerable efficiency and successful reprogramming of the TME was detectable [311] which was also observed in other preclinical studies [137]. Another “normalization” or “reprogramming” strategy might be the modulation of the patient’s microbiome. As outlined above, the microbiome substantially influences the efficacy of chemotherapeutic drugs in PDAC-patients [11,19,20] as well as the response to immune checkpoint inhibition [298]. Additionally, restoration of a tumor-restraining microbiome by application of distinct microbiome species, e.g., such as FMT, may not only overcome drug resistances, but also help to boost the patient’s immune system to induce a long-term control of the tumor burden [11,20].

Overall, the above-mentioned studies and continuously elaborated novel concepts have essentially broadened our understanding of this challenging disease. However, additional studies are needed to further dissect the dynamics and heterogeneity of the TME regarding the tumor-promoting and tumor-restraining functions in order to translate this knowledge into personalized therapeutic concepts to improve the prognosis of PDAC patients. Consideration of this heterogeneity and its impact on PDAC development and therapy responses is enabled by collecting and comprehensively analyzing tumor, stroma and the patient’s microbiome to generate an integrated tumor subtype. This identified subtype allows stratification of each patient for the individualized and most effective therapy (Figure 2).

## 5. Conclusions

Extensive preclinical and clinical evaluation of PDAC has resulted in detailed characterization of its complexity on multiple levels. In this article, we summarized how PDAC biology is determined by a plethora of cell types and microbiota shaping the TME and their context-dependent pro- or anti-tumorigenic characteristics. This challenging level of complexity is further increased by countless interactions between the involved cell types. In the future, increased efforts will be needed to translate these insights into TME heterogeneity into clinically meaningful benefits for PDAC patients. To this end, we propose comprehensive analysis of patient material (tumor, tumor stroma and microbiome) and the development of integrated tumor subtypes based on this data base. Respective integrated tumor subtypes may then serve for informed patient stratification in clinical trials and truly individualized therapeutic decisions.

## Figures and Tables

**Figure 1 cancers-13-04932-f001:**
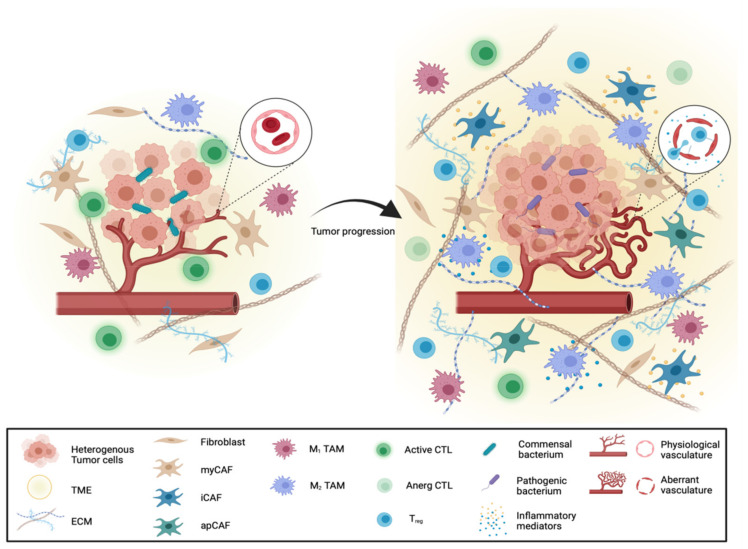
Pancreatic cancer is characterized by a high tumor and stroma heterogeneity. During pancreatic tumorigenesis, tumor cells acquire multiple alterations resulting in a heterogeneous pool of tumor cell clones. Along with the evolution of these divergent tumor cell clones, the tumor microenvironment (TME) also undergoes fundamental alterations involving, (i) activation of fibroblasts and transdifferentiation into carcinoma associated fibroblasts (CAFs) with different phenotypes (myCAF, iCAF, apCAF), (ii) alteration of the extent and composition of the extracellular matrix (ECM), (iii) enrichment of immunosuppressive and tumor-promoting immune cells e.g., tumor-associated macrophages with different phenotypes (M1 TAM versus M2 TAM), anergy cytotoxic T cells (CTL) and regulatory T cells (Treg), (iv) enrichment of pathological bacteria and extrusion of commensal bacteria, (v) elevation of inflammatory mediators and (vi) formation of new and modified blood vessels. Overall, this TME heterogeneity is an essential driver of PDAC development and progression. Figure created with BioRender.com.

**Figure 2 cancers-13-04932-f002:**
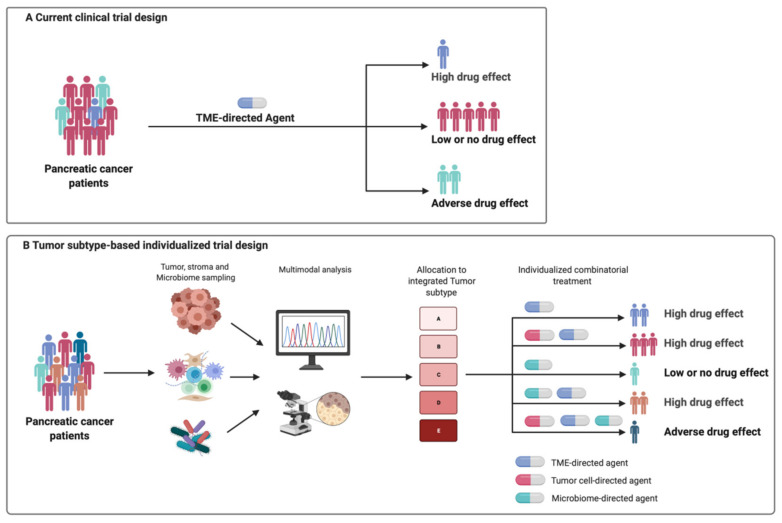
Current and future therapeutic concepts for the treatment of pancreatic cancer patients. (**A**) In current clinical trials, pancreatic cancer patients are not stratified according to distinct tumor and stroma characteristics. Accordingly, the given therapy exerts an anti-tumor effect only in few patients. However, in the majority of patients this strategy fails or has even adverse effects. (**B**) A tumor subtype-based individualized trial design implies sampling of the patient’s tumor, stroma and microbiome prior to treatment and multimodal analysis of these samples will generate an integrated tumor subtype for each patient. Considering these tumor, stroma and microbiome characteristics, the identified subtype allows stratification of pancreatic cancer patients for the treatment strategy with the best predicted outcome. Figure created with BioRender.com.

**Table 2 cancers-13-04932-t002:** Overview of the cited therapeutic approaches targeting macrophages.

Study System	Targeting Strategy	Reference
**Preclinical studies**		
KPC mice	CD40 agonist treatment by CP-870,893	[177]
KPC mice	CD11b agonist treatment by ADH-503	[183]
C57BL/6 mice including Batf3 KO, CD40 KO, MyD88 KO, STING KO and IFNAR KO	CD40 agonist treatment by FGK45 combined with anti-PD-1 by RMP1-14 and/or anti-CTLA-4 by 9H10	[179]
**Clinical studies**		
Phase1 clinical trial	CD40 agonist treatment by Selicrelumab	[178]
Phase 1b clinical trial	CD40 agonist treatment by Sotigalimab in monotherapy or in combination with PD-1 blockade by Nivolumab	[180]
Phase 2 clinical trial	CD40 agonist treatment by Sotigalimab in monotherapy or in combination with PD-1 blockade by Nivolumab	[181]
Phase 1b/2 clinical trial	CD40 agonist treatment by Sotigalimab in combination with PD-1 blockade (Nivolumab) + Gemcitabine + Nab-Paclitaxel, or Sotigalimab + Gemcitabine + Nab-Paclitaxel.	NCT03214250
Phase 2 clinical trial	Addition of recombinant fms-like tyrosine kinase 3 (Flt-3) ligand (CDX-301) to the CD40 agonistic antibody (CDX-1140)	NCT04536077
Phase 2 clinical trial	modified FOLFIRINOX (mFFX) combined with ABBV-927 with or without Budigalimab	NCT04807972
Phase 1b/2 clinical trial	CD40 agonist Mitazalimab in combination with modified FOLFIRINOX	NCT04888312

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
