# Peer review of "The Heterogeneity of the Tumor Microenvironment as Essential Determinant of Development, Progression and Therapy Response of Pancreatic Cancer"

_cancers, 2021, doi:10.3390/cancers13194932_

Round 1
Reviewer 1 Report
The authors do a very nice job summarizing the latest findings on the role of tumor microenvironment in the development of pancreatic cancer. The review is very well written and rounded. Overall, the opinions presented are balanced and cite relevant recent studies. There are some areas that could be discussed more thoroughly, such as:
-It would be helpful for the reader to have a more inclusive discussion of additional cell types that have been implicated in pathogenesis of PDAC, such as dendritic cells, B cells and definitely MDSCs.
-There are a couple of really excellent studies that could be mentioned as relevant to macrophage biology and targeting opportunities, such as: https://pubmed.ncbi.nlm.nih.gov/30827862/ and https://pubmed.ncbi.nlm.nih.gov/31270275/
-the mention that PDAC is of ductal origin could be better discussed, as it quite likely can originate from either ductal or acinar compartments.
Author Response
Reviewer 1:
The authors do a very nice job summarizing the latest findings on the role of tumor microenvironment in the development of pancreatic cancer. The review is very well written and rounded. Overall, the opinions presented are balanced and cite relevant recent studies. There are some areas that could be discussed more thoroughly, such as:
Request 1: “It would be helpful for the reader to have a more inclusive discussion of additional cell types that have been implicated in pathogenesis of PDAC, such as dendritic cells, B cells and definitely MDSCs.”
Answer 1: We thank the reviewer for critical reading our manuscript and his commendation. As requested, we have incorporated some information on the respective cell lines (page 9 and 12). However, as these cell populations are not in the focus of the review, we have not extensively discussed the recent knowledge on these cells.
Request 2: “There are a couple of really excellent studies that could be mentioned as relevant to macrophage biology and targeting opportunities, such as: https://pubmed.ncbi.nlm.nih.gov/30827862/ and https://pubmed.ncbi.nlm.nih.gov/31270275/”
Answer 2: We thank the reviewer for the helpful advice and have incorporated both publications (see page 10+11).
Request 3: “the mention that PDAC is of ductal origin could be better discussed, as it quite likely can originate from either ductal or acinar compartments. “
Answer 3: We have incorporated a respective notion in section 1 on page 2.
Reviewer 2 Report
This is a well-written review covering contemporary knowledge on the tumor and stroma compartment in PDAC. The structure is clear; the authors describe the heterogeneity of cancer associated fibroblasts (CAF), macrophages, T cells, endothelial cells and microbiome. Each paragraph is then accompanied by approaches for modulation of CAF, macrophages, T cells, endothelial cells/angiogenesis or microbiome. Finally, there is a paragraph with a summary and an outlook.
- In general, the review would be enhanced by consistent indications which evidence is form mouse studies and which from human PDAC studies.
- It will be helpful to summarize the discussed data on the heterogeneity in tumor cell compartment as well as approaches to modulate fibroblasts and macrophages as tables.
- For Tregs it has been shown that the depletion can be detrimental because of stroma modification (DOI: 10.1158/2159-8290.CD-19-0958). This should be discussed.
Author Response
Reviewer 2:
This is a well-written review covering contemporary knowledge on the tumor and stroma compartment in PDAC. The structure is clear; the authors describe the heterogeneity of cancer associated fibroblasts (CAF), macrophages, T cells, endothelial cells and microbiome. Each paragraph is then accompanied by approaches for modulation of CAF, macrophages, T cells, endothelial cells/angiogenesis or microbiome. Finally, there is a paragraph with a summary and an outlook.
Request 1: “In general, the review would be enhanced by consistent indications which evidence is form mouse studies and which from human PDAC studies.”
Answer 1: We thank the reviewer for critical reading our manuscript and his commendation. If not already performed, we have more consistently indicated whether the findings were made in human or murine cells/tissues throughout the paper.
Request 2: “It will be helpful to summarize the discussed data on the heterogeneity in tumor cell compartment as well as approaches to modulate fibroblasts and macrophages as tables.”
Answer 2: Since this review focuses on the stromal cell heterogeneity we do not wish to further extent the section on the tumor cell heterogeneity, As requested, we have summarized the cited studies on CAF therapeutic approaches in the new Table 1 (see page 8) and the macrophage targeting approaches in the new Table 2 (page 12).
Request 3: “For Tregs it has been shown that the depletion can be detrimental because of stroma modification (DOI: 10.1158/2159-8290.CD-19-0958). This should be discussed.”
Answer 3: As requested we have discussed this finding on page 15.